# A Novel Cell Map Representation for Weakly Supervised Prediction of ER & PR Status from H&E WSIs

**Abstract.** Digital pathology opens new pathways for computational algorithms to play a significant role in the prognosis, diagnosis, and analysis of cancer. However, handling large whole slide images (WSIs) is a vital challenge that these algorithms encounter. In this paper, we propose a novel technique that creates a compressed representation of histology images. This representation is composed of cellular maps and compresses the WSIs while keeping relevant information at hand including the spatial relationships between cells. The compression technique is used to predict the status of ER & PR expressions from H&E WSIs. Our results show that the proposed compression technique can improve the prediction performance by 11-26%.

**Keywords:** Computational Pathology · ER/PR prediction · Compressed representations · Breast cancer.

## 1  Introduction

Processing the histopathological whole slide images (WSIs) is a challenging task due to their multi-gigapixel sizes. A naïve solution to handle these images in machine learning (ML) models is down-sampling them to a small size image. However, down-sampling destroys high amount of contextual information and may lead to poor results. In practice, analysing the WSIs at cellular level is essential to understand the tumour micro-environment or TME [3]. An alternative solution for handling these images in the ML models is by splitting them into small image patches of manageable size. Despite the fact that this solution is popular, the relation between different patches and their surrounding contextual information is lost.

In [21], the authors compress the image using self-supervised approaches where a CNN is trained in a self-supervised manner and then the feature maps that are generated by the CNN are used as the compressed representation of a given image. Streaming convolution and gradient check-pointing used by [15] is a technique which reduces the memory consumption at the cost of increasing computations. In this approach, large images at multiple branches are fed into the model. Multiple Instance Learning (MIL) approaches presented in the computational pathology literature have also been widely used recently for dealing with weakly-supervised tasks by treating WSIs as bags of images [2]. However, these approaches lack interpretability and do not account for information about

nuclei positions and categories. Moreover, most of them ignore the relational information within the WSIs.

In this paper, we propose a novel representation (*CellMaps*) that not only reduces the image size, but also represents the WSIs based on the cellular density. The aim of *CellMaps* is to keep all the relevant information intact while reducing the image size. This representation keeps the cellular level details, besides capturing spatial information from the original image. In our experiments, we show that representing the cellular density along with the contextual information improves the final predictions. *CellMaps* can be used for compressing images at different desired levels, depending on the task and available computational capacity.

We employ our approach for the evaluation of Oestrogen Receptor (ER) and Progesterone Receptor (PR) expression which are essential prognostic and predictive factors for breast cancer (BC) patients [7]. Based on the level of positivity of ER/PR, chemotherapy or endocrine therapy is often determined [12].

BC tissues are stained with Immunohistochemistry (IHC) biomarkers in routine clinical practice, followed by a visual assessment by pathologists estimating ER/PR expression distribution across all the tumour tissues [13]. This practice poses two main challenges: First, the IHC makers are costly and laborious. Second, it may face lack of reproducibility as it relies on visual analysis by the pathologists. Therefore, objective automated techniques that can overcome these challenges are in high demand, specially for predicting ER/PR expressions which are costly and highly subjective tasks.

Recent studies proposed techniques and automated tools predicting the hormonal expression in BC tissue. Several studies conduct experiments on tissue micro-array (TMA) core images [20, 17, 18]. On the other hand, some studies have been conducted at the WSI level. Naik *et al.* [14] reported the state-of-art results with an AUC of 0.92, but the approach was trained in a supervised manner on detailed regions of interest (ROIs) that were annotated by pathologists. Rawat *et al.* [16] perform a study on a large cohort. Nevertheless, their approach neither includes the entire extracted patches from the WSI nor set clear exclusion criteria. The approach randomly selected patches that may face reproducibility issues. Likewise, Lu *et al.* [11] proposed the Slide Graph technique, which is based on graph convolutional neural network (GCN). Their technique processes the entire WSI to predict PR and Her2 status, with AUCs of 0.73 and 0.62, respectively. To the best of our knowledge, ours is the first study to predict ER as well as PR by including the entire WSIs with clear exclusion criteria, which is more relevant clinically.

**The main contributions of this paper are as follows:**

1. We present a novel compression technique (*CellMaps*) that represents image patches by the cellular density, while keeping the spatial information intact.
2. We present a pipeline predicting the status of ER and PR in TCGA BC cohort using *CellMaps* of WSIs, with specific and clear exclusion criteria.
3. We show that the prediction performance is improved when using the compressed images, compared to using the raw H&E image.

The paper is organised as follows: In Section 2, we describe the materials and methods, followed by a discussion of our results in Section 3. We conclude the paper in Section 4.

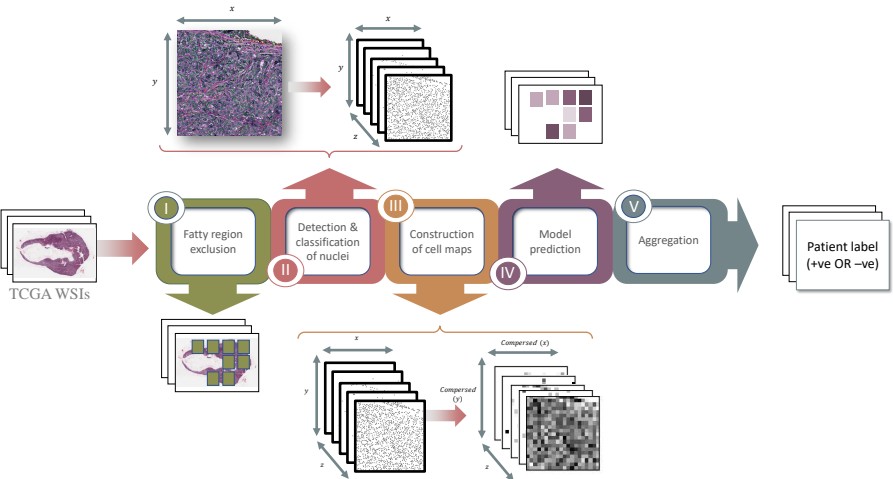

**Fig. 1.** An overview of the proposed pipeline. Step III generates the *CellMaps* compressed representation for every patch that is then fed into a model for patch level prediction.

## 2    Materials and Methods

### 2.1    Materials

We gathered 356 WSIs from The Cancer Genome Atlas (*TCGA*), multi-center data [8]. The collected cohort is a subset of diagnostic cases of TCGA. Additionally, the corresponding molecular status (i.e., ER and PR status) of the cohort were also collected.

### 2.2    The Proposed Methodology

Our pipeline consists of five main phases. We first exclude fatty regions from the study as they relatively provide less cellular information, compared to other tissue types. Afterwards, we extract the location of five different cell types in a WSI in order to build a cell map. The representation of the cell map is shown in Fig 2, where each cell type is represented in a single layer in the cell map image. We then extract a fixed tile size from each WSI in our study so that we maintain a fixed input size for our model. Next, we utilised the widely-used `Resnet18` to classify the cell map representation of each extracted tile into +ve or -ve based on the patient-level label from TCGA. We train two separate models for each task (i.e., ER and PR classification). Lastly, we aggregate the tile results at the patient level.

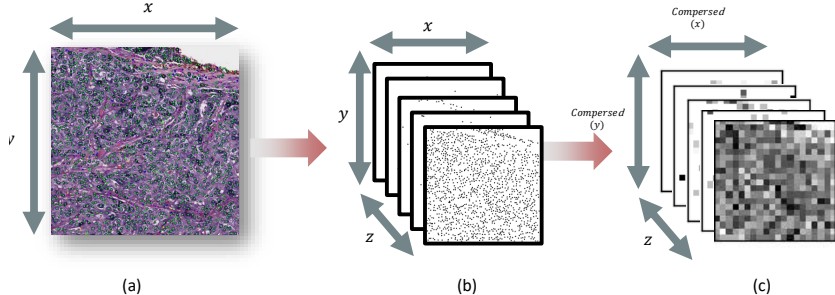

**Fig. 2.** An example of how a tile and is compressed. (a) is an image tile. (b) shows a set of images after extracting the nuclei locations and types. (c) is the *CellMaps* representation of the tile.

**(I) Fatty regions exclusion.** During tile extraction, we build a mask identifying eleven tissue types, one of which is adipose (i.e., fatty) tissue. The `Resnet18` model was pre-trained with `Kather100k` dataset [9]. A tile is excluded if the majority of its tissue (50% or more) contains fatty regions.

**(II) Detection and classification of nuclei.** To find the locations and types of cells in a given image, we leveraged *HoVer-Net* [5]. The *HoVer-Net* model was trained with the *PanNuke* dataset [4] consisting of more than 200k labelled nuclei from 19 different tissue types.

Having identified the location and type of each nucleus presented in a given tile, we generate a preliminary cell map representation. This representation has three dimensions: $x$, $y$ and $z$, see Fig 2 (b). The $x$ and $y$ dimensions of the preliminary representation are the same as the $x$ and $y$ of the original tile shape, presented in Fig 2 (a), whereas, the $z$ dimension corresponds to the number of cell types, each of which is represented in a single layer. In our experiment, we have five different cell types (neoplastic, inflammatory, connective, dead, and non-neoplastic), so the representation has five different layers, i.e., $z = 5$.

**(III) Construction of cell maps.** The aim of this stage is to compress the extracted tiles into a smaller size that the classification model (in stage (IV) of our pipeline) can handle. The representation (as in Fig 2 (c)) shows the cellular density of a given tile for many cell types. It has three dimensions: $compressed(x)$, $compressed(y)$ and $z$.

*CellMaps* can compress a given image/tile into a smaller size but keeping cellular level details, along with their spatial maps to keep the information about cell-cell interactions. The technique utilises average filtering, which computes the number of cells presented in a pre-defined size of the average filter. Then each pixel in the representation image is assigned a ratio (between 0 and 1) based on the cell density of the original image corresponding to a window filter size. The larger the filter size, the smaller the *CellMaps* representation. Fig 3 shows an example of tile and its representations with various sizes using *CellMaps*.

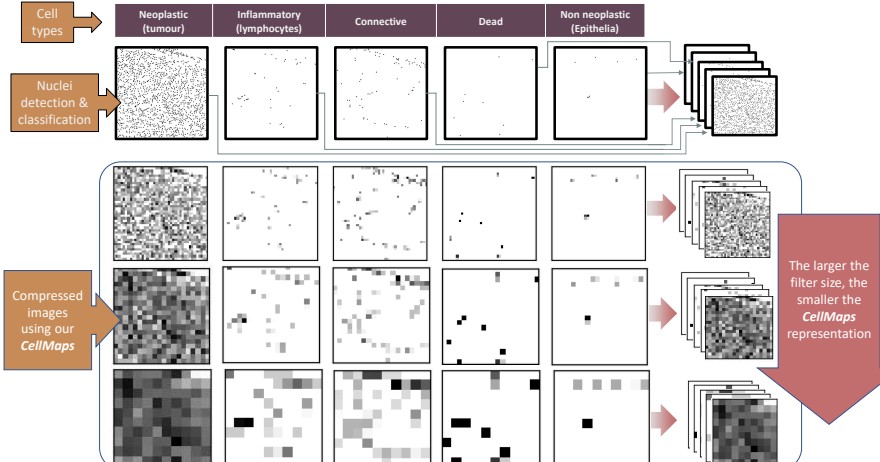

**Fig. 3.** An example of tile and its representations with various filter sizes (i.e., size of avg filter) using *CellMaps*. The filter sizes are 160, 320, and 480 pixels, from the top row to the bottom. The images from each cell type is concatenating to build the compression representation, as shown in the first row.

The *CellMaps* assists ML models in handling the entire WSIs (or large tile images) without losing the detailed cellular level information. In our experiment, we choose to deal with large tiles in order to overcome the WSI size variations in TCGA data.

**(IV) Model prediction.** `Resnet18` [6] is employed for binary classification of the *CellMaps* representations of the extracted tiles into +ve or -ve. The tiles' ground truth labels are based on the patient level ER and PR status. Two separate models are being trained (i.e., one for ER and the other for PR status).

**(V) Aggregation.** The different tiles belonging to one WSI are aggregated to find the ER/PR status at the patient level. We apply majority voting (MV) of the tile predictions. A WSI is considered positive if more than 50% of tiles are predicted positive by the models in stage IV.

## 3    Experiments and Results

### 3.1    Datasets

The TCGA collected cases are randomly divided into three datasets: (1) training (50% of cohort), (2) validation (25%), and (3) testing (25%). Table 1 shows the datasets used for both experiments (i.e., ER & PR prediction). We maintained the same distribution of positive and negative classes among all datasets.

Our experiment can be designed at compressing each WSI as a single *CellMaps* representation, i.e., each WSI is compressed and then fed into the model as one single image. However, this is challenging due to size variations in WSIs. To overcome this challenge, the experiment was designed at dividing the WSIs

**Table 1.** The training, validation, and testing datasets.

| Molecular | ER | | | PR | | |
|---|---|---|---|---|---|---|
| Classes | (+)ve | (-)ve | Total cases | (+)ve | (-)ve | Total cases |
| Training | 139 | 39 | 178 | 120 | 57 | 177 |
| Validation | 69 | 19 | 88 | 60 | 29 | 89 |
| Testing | 70 | 20 | 90 | 60 | 29 | 89 |
| Entire | 278 | 78 | 356 | 240 | 115 | 355 |

into tiles. Nevertheless, we chose to extract large tiles, sized $9600 \times 9600$ at $40\times$ magnification, so that the model captures significant contextual information during training. Table 2 presents the number of tiles extracted for each dataset.

**Table 2.** Number of extracted tiles for each dataset.

| Molecular | Training | Testing | Validation | Entire dataset |
|---|---|---|---|---|
| ER | $8,026$ | $3,730$ | $4,387$ | $16,143$ |
| PR | $8,212$ | $3,922$ | $3,929$ | $16,063$ |

### 3.2 Experimental setup

`Resnet18` was chosen as a binary classifier for its robustness, reliability, and wide usage in medical imaging applications [1, 10, 19]. The model was trained for 100 epochs, and we added a dropout layer (with the configuration of 0.2) before the last fully connected layer to avoid over-fitting. The learning rate was initialised with a value of 0.01, and a scheduler was implemented to decrease the learning rate after each epoch such that the model becomes more stable at the later training stages. The threshold determining the tile positivity are set based on the best performance on the validation dataset.

### 3.3 Evaluation of classification performance

*The significance of excluding fatty region.* We conducted several experiments, one of which was conducted without filtering out the fatty regions (FRs) of WSIs in order to draw a fair evaluation of adding this stage to our proposed pipeline. Table 3 shows the performance of two experiments: (1) **with** and (2) **without** the FR exclusion. The four columns to the right side present the performance **without** excluding FRs, while the columns to the left side present the performance **with** the exclusion of FRs.

Fig 4 presents the area under the receiver operating characteristic (AUC-ROC) for validation and testing datasets for both ER and PR experiments. The bottom row presents the performance when excluding the FRs, whereas the

**Table 3.** The performance of both experiments, **with** and **without** excluding FRs.

| Experiment | **With** excluding FRs | | | | **Without** excluding FRs | | | |
|---|---|---|---|---|---|---|---|---|
| Molecular | ER | | PR | | ER | | PR | |
| Dataset | Validation | Testing | Validation | Testing | Validation | Testing | Validation | Testing |
| Accuracy | 0.83 | 0.81 | 0.69 | 0.66 | 0.84 | 0.78 | 0.64 | 0.674 |
| Precision | 0.88 | 0.84 | 0.79 | 0.72 | 0.83 | 0.80 | 0.7 | 0.72 |
| Recall | 0.91 | 0.93 | 0.73 | 0.8 | 1.00 | 0.96 | 0.8 | 0.85 |
| $f_1$-score | 0.89 | 0.88 | 0.76 | 0.76 | 0.91 | 0.87 | 0.75 | 0.78 |
| **AUC** | **0.83** | **0.77** | **0.73** | **0.65** | **0.83** | **0.72** | **0.69** | **0.61** |

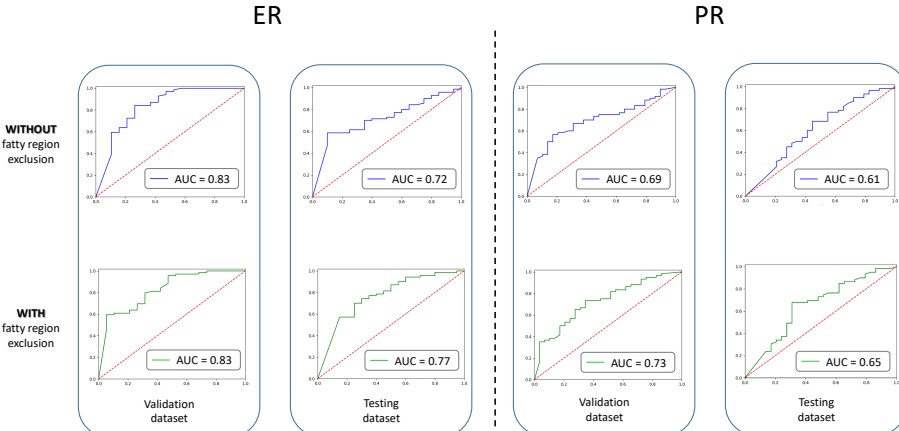

**Fig. 4.** The ROCs of the pipeline before and after excluding the FRs

top row shows the experiment results without filtering the FRs. We can see a noticeable improvement when excluding the FRs, with 4-5% in the *AUCs* of the testing datasets for ER and PR predictions.

**Table 4.** The performance when using raw H&E images.

| Molecular | ER | | PR | |
|---|---|---|---|---|
| Datasets | Validation | Testing | Validation | Testing |
| Accuracy | 0.7 | 0.63 | 0.63 | 0.62 |
| Precision | 0.86 | 0.77 | 0.82 | 0.73 |
| Recall | 0.73 | 0.75 | 0.66 | 0.67 |
| $f_1$-score | 0.79 | 0.76 | 0.73 | 0.7 |
| **AUC** | **0.62** | **0.51** | **0.72** | **0.54** |

*Comparing CellMaps vs raw (H&E) images.* To examine the relevance of our *CellMaps* technique, another experiment was also conducted. We compare the

performance using the raw H&E images (instead of the compressed images) with the same proposed pipeline, including the filtration of the FRs. The aggregation of the raw images are also based on the majority vote. Table 4 shows a considerable drop of $(11 - 26\%)$ in the AUCs for the prediction of ER and PR.

*The sensitivity of filter size.* Different filter sizes were implemented so as to evaluate its sensitivity. Table 5 presents four different filter sizes: 160, 320, 480, and 640 pixels at $40\times$ magnification. The Table does not show a major drop/jump in the performance (i.e., AUCs of testing datasets) when changing the filter size of the compression technique (*CellMaps*). These results indicate that the model captures the cell density of a given image, regardless of the level of details. Hence, using our *CellMaps* input images can be compressed to the desired size that the ML model can handle without destructing the performance.

**Table 5.** The performance when changing the filter size.

| Molecular | Performance metrics | filter size (in pixels at $40\times$ magnification) | | | |
|---|---|---|---|---|---|
| | | 160 | 320 | 480 | 640 |
| ER | Accuracy | 0.72 | 0.74 | 0.72 | 0.81 |
| | Precision | 0.85 | 0.85 | 0.83 | 0.84 |
| | Recall | 0.79 | 0.81 | 0.81 | 0.93 |
| | $f_1$-score | 0.81 | 0.83 | 0.82 | 0.88 |
| | **AUC** | **0.72** | **0.75** | **0.71** | **0.77** |
| PR | Accuracy | 0.68 | 0.64 | 0.60 | 0.66 |
| | Precision | 0.71 | 0.71 | 0.75 | 0.72 |
| | Recall | 0.88 | 0.78 | 0.61 | 0.80 |
| | $f_1$-score | 0.79 | 0.74 | 0.67 | 0.76 |
| | **AUC** | **0.65** | **0.65** | **0.61** | **0.65** |

## 4 Conclusions and Future Work

Computational algorithms encounter a crucial challenge while processing multi-gigapixel histology images. Most machine learning or ML models cannot handle such large sizes, requiring a division into small patches. Instead, one may design an algorithm handling large images but down-sampled. ML models usually capture patterns from the detailed information in histology images.

In this paper, we presented a novel compression technique, the *CellMaps*. Our technique is based on average-filtering, yet efficient as it keeps spatial information intact which is useful for analysing the TME. It does not only reduce the image size, but also represents the cellular density, which can improve the prediction performance, as our results show. In future, we will explore the efficacy of this representation for other tasks in computational pathology.

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
