# OpenReview forum: "A Novel Cell Map Representation for Weakly Supervised Prediction of ER and PR Status from H&E WSIs"
_MICCAI.org/2021/Workshop/COMPAY — COMPAY 2021_

### Official Review · Reviewer_B5yB · 2021-08-17
**good initial results**

**Rating:** 6
**Confidence:** 3

**Review:**

The authors propose an approach based on something akin to celltype-count-averaging: 1) exclude fat 2) detect cell types in in tiles 3) compress tiles by averaging per cell type with a certain filter size 4) classify tiles, 5) classify slides/cases by major vote. For 2 a (pretrained) hovernet is used. The authors achieve test aucs of 0.72-0.77 (which is not bad for a task as difficult as this one) and show that processing the raw slides yields only AUCs around 0.5.
This is good initial work, for an extended work (or improving this paper) I would suggest to: a) show all the auc-results in a single table, this will make the results much more clear. b) analyze what the network looks at (e.g. gradcam). Currently, it's not clear why the network makes a prediction - with only 90 test cases an auc of ~0.6 might be achieved just by chance. c) this depends on b, but its also unclear whether all the detected cell types (by the hovernet) are necessary for the prediction. d) finally, the results depend a lot on the second step - the pretrained hovernet. Is it precise enough for the tcga data, is it biased towards predicting normal cells as tumor or vice versa?

---

### Official Review · Reviewer_V6wT · 2021-08-20
**Well described and experimentally solid but lacks evaluation and motivation**

**Rating:** 5
**Confidence:** 4

**Review:**

The authors propose a technique for creating compressed representations of histology images by first generating multi-class nuclei detection maps and then compressing these maps by average filtering. This technique is then applied to the classification problem of estrogen and progesterone receptor status prediction. The authors perform experiments where they compare the classification performance given different compression rates, the inclusion & exclusion of patches containing fatty regions and the usage of H&E patches versus their nuclei detection maps.

**Major comments:**

The design of the proposed technique is described well conceptually, but I believe the motivation for its design, as stated in the introduction, is flawed.

- The authors note that techniques focusing on gigapixel images commonly use patch-based approaches that ignore the relational information within WSIs; however, I would argue that via the two step process of multi-class nuclei detection on tiles (effectively just larger patches) and then aggregating the tile-level predictions using majority voting, you don't necessarily take this relational information into account either. This is partly offset by the fact the authors use large tiles (9600x9600 at 40x), but this does not solve the issue.
- Two of the methods that the authors mention (neural image compression and streaming) do in fact consider the full WSI (a compressed representation in the case of NIC).
- The authors note the mentioned approaches lack interpretability, but do not address this issue themselves.

The performed experiments are relevant and well-designed, but I believe discussion is lacking on some key points, e.g. why the CellMaps technique outperforms using the raw H&E images, especially in light of Naik *et al.*, who achieve a much higher AUC using only H&E images. Moreover, I believe more quantitative comparisons to similar WSI-level classification techniques (such as streaming, NIC or MIL) are warranted to prove the potential of the method, although I understand their lack given the amount of work required and the nature of workshop papers.

**Minor comments:**

- Can the authors comment on why they think estrogen receptor status is easier to predict than progesterone receptor status, as the ER class consistently scores higher than PR?
- I think it makes more sense to use bolding in tables to highlight the best performing metric, instead of highlighting the AUC values.
- I think having the in-line citation numbers be ordered according to their appearance in the text would increase legibility.
- A final pass for small grammar/spelling mistakes would help the legibility of the paper:
    - 'First, the IHC ma[r]kers...' (P2)
    - 'Fig 3 shows an example of [a] tile...' (P3)
    - 'The significance of excluding fatty region[s]' (P6)
    - 'The ROC [curves] of the pipeline before and after excluding the FRs' (P7)

---

### Decision · Program_Chairs · 2021-08-25

Accept